# Antibacterial effects of *thyme* oil loaded solid lipid and chitosan nano-carriers against *Salmonella Typhimurium* and *Escherichia coli* as food preservatives

Amirhosein Shabgoo Monsef[1], Mehran Nemattalab[1,2], Shirin Parvinroo[3], Zahra Hesari[1] *

1 Department of Pharmaceutics, School of Pharmacy, Guilan University of Medical Sciences, Rasht, Iran,
2 Department of Microbiology, School of Medicine, Guilan University of Medical Sciences, Rasht, Iran,
3 Department of Pharmacognosy, School of Pharmacy, Guilan University of Medical Sciences, Rasht, Iran

* z.hesari@gmail.com, z.hesari@gums.ac.ir

## Abstract

### Objectives

*Escherichia coli* and *Salmonella Typhimurium* are frequent causes of foodborne illness affecting many people annually. In order to develop natural antimicrobial agents against these microorganisms, thyme oil (TO) was considered as active antibacterial ingredient. TO contains various bioactive compounds that exhibit antimicrobial properties. To increase the antibacterial effects and stability of thyme oil, two promising carrier systems, solid lipid nano-particles (SLN) and chitosan nanoparticles have been fabricated in this study.

### Methods

Nanoparticles were made using natural-based lipids and polymers by a probe sonication method. They were characterized using infrared spectrometry (FTIR), transmission electron microscopy (TEM), particle size, cytotoxicity, etc. Antibacterial effects of TO, thyme oil loaded in SLN (TO-SLN) and thyme oil loaded in chitosan nanoparticle (TO-CH) was evaluated against *E. coli* and *S. typhimurium* using Minimum inhibitory/bactericidal concentrations (MIC/MBC) determination. Encapsulation efficiency (EE%) and drug release profile were also studied *in vitro*.

### Results

TEM analysis revealed spherical/ovoid-shaped particles with clear edges. TO-SLN had an average size of 42.47nm, while TO-CH had an average size of 144.8nm. The Encapsulation efficiency of TO–CH and TO–SLN nanoparticles were about 81.6±1% and 73.4±1%, respectively. Results indicated 92% cumulative release in TO-CH in comparison with 88%

**Data Availability Statement:** All relevant data are within the paper and its Supporting Information files.

**Funding:** The author(s) received no specific funding for this work.

**Competing interests:** The authors have declared that no competing interests exist.

in TO-SLN in 72 h. MIC against *E. coli* and *S. typhimurium* for TO-CH, TO-SLN, and pure TO were 4 and 1.5 μg/mL, 60 and 40 μg/mL, and 180–150 μg/mL, respectively.

## Conclusion

Nanoencapsulation of thyme oil significantly potentiated its antimicrobial effects. TO-CH exhibited a significantly higher antibacterial effect compared to TO-SLN (6-fold) and pure thyme oil (more than 10-fold).

## 1. Introduction

Foodborne illnesses are among the biggest global health challenges that is expanding for both human and animal populations. Every year, a large number of people are affected by consuming food contaminated with pathogenic *Salmonella* and *Escherichia coli* (*E. coli*) [1]. *E. coli* is a Gram-negative bacterium that is rod-shaped, facultative anaerobic, non-spore-forming and belongs to the *Enterobacteriaceae* family [2]. These bacteria are present as a component of the normal microbiota in the human gastrointestinal tract [3]. However, certain strains of *E. coli* are pathogenic which can be classified into three categories based on their pathogenicity: commensal strains, intestinal pathogenic *E. coli*, and extra-intestinal pathogenic *E. coli* [4]. *E. coli* causes food poisoning and affects the food-processing industry, health sectors, and agriculture [5]. Also, *E. coli* is a critical threat to children's health because it is one of the main bacterial species implicated in the occurrence of diarrhea and is the second biggest killer of children under 5 years old after pneumonia [6].

Salmonella is responsible for the most common food-borne disease in poultry and poultry products such as meat (pork, chicken, fish, and beef), vegetables, milk, eggs, fresh fruits, and cheese, and is the main cause of diarrheal diseases and human salmonellosis [7–9]. *Salmonella* is classified as a gram-negative, flagellated, facultative anaerobic bacillus from Enterobacteriaceae family. Salmonellosis is a zoonotic disease which causes severe pathogenic complications [10]. The United States Department of Agriculture has put the control of salmonella in poultry in a high priority due to the serious public health ramifications [11]. Because it forms biofilms, *Salmonella* resist against disinfection efforts and can detect, adapt to, and survive in stressful environmental conditions [7].

There has been a significant clinical interest about natural medicine and the widespread usage of herbal products in recent years [12]. On the other hand, the emergence of newer diseases and the increasing multi-drug resistant microorganisms require the urgent development of novel drugs. Essential oils (EOs) are among the most powerful plant derivatives that are considered as a renewable source for discovering new antimicrobial agents [13, 14]. EOs are hydrophobic and cause bacterial cell membranes to rupture and leak their cell contents into the surrounding environment [15]. Hence, the role of EOs as natural food preservatives has received attention due to their combinational benefits as antioxidant and antimicrobial compounds [16]. *Thymus vulgaris* Linn. from the *Lamiaceae* family, is a herbaceous plant [17] that is found in southern European and mediterranean region [18, 19]. Thyme has historically been employed in traditional medicine for its analgesic, antibacterial, anthelmintic, antidiarrheal, and carminative properties. Contemporary pharmacological investigations have unveiled the extensive range of biological features exhibited by the subject, including antibacterial, antinociceptive, spasmolytic, and anti-inflammatory activities [20]. Thyme essential oils

comprise many bioactive compounds, including thymol, flavonoids, terpenoids, carvacrol, etc., with antitumor, antiviral, antioxidant, antimicrobial, and hypoglycemic effects [21–23].

Chitosan is a naturally occurring polymer derived from chitin, a compound composed of N-acetyl-d-glucosamine units connected by β-(1,4)-glycosidic bonds [24, 25]. It offers various biochemical advantages including antioxidant, antibacterial, remarkable biocompatibility, and film-forming properties, which have made it a choice to be widely used in food storage, food packaging, and as a food additive [26]. For example, chitosan nanoparticles encapsulating thyme oil and oregano oil presented an increased bactericidal activity against foodborne pathogens (*S. aureus*, *E. coli*, *L. monocytogenes*) [27]. I addition, thyme oil loaded chitosan nanoparticles aa nano-capsules presented an inhibitory activity against six foodborne bacteria [28]. Also, in recent decades, chitosan nanoparticles have been widely used as drug delivery carriers due to their ability to maintain the stability and potentiate the biological effects of loaded drug molecules [29, 30].

Solid lipid nanoparticles are carrier systems with a lipid base and sizes in the range of 30 to 1,000 nm [31]. SLNs are systems with prominent properties, such as the possibility of high compatibility and drug loading capacity, drug targeting and controlled release, the capacity to safeguard unstable drugs from degradation and the ease of production on a large scale [32]. They consist of biodegradable lipids that are solid in the form of aqueous colloidal dispersions, and possess the combined benefits of nano-emulsions and liposomes at the same time [33]. Nano-lipid carriers consisted of thyme/pennyroyal essential oils loaded by saffron extract revealed antimicrobial effect against *S.aureus*, *E. coli*, and *P. aeruginosa* while saffron extract itself, did not present antimicrobial effect [34].

However, numerous studies have confirmed the antimicrobial properties of thyme, and it has been incorporated into various nano-delivery platforms. To our knowledge, there is no study comparing chitosan with SLN as carrier systems for the antibacterial effects of thyme oil, yet. Hence, in current study, SLN and chitosan nanoparticles have been developed as thyme oil nanocarriers to potentially enhance their antimicrobial effects against *S. typhimurium* (ATCC 14028) and *E. coli* (ATCC 25922) as food-contaminating bacteria. All incorporated excipients are natural, non-toxic, and biocompatible for human use.

## 2. Materials and methods

### 2.1. Ethics approval

All procedures in this study followed ethical standards (IR.GUMS.REC.1402.457) released by the Ethical Review Committee of Guilan University of Medical Sciences, Rasht, Iran. Written consent was not required.

### 2.2. Preparation of solid lipid nanoparticles containing thyme oil

In this study, TO-SLN was constructed by the emulsification ultrasonic-homogenization method. Briefly, 100 mg of cholesterol (Sigma-Aldrich, Germany) and 100 mg of Lecithin (Merck, Darmstadt, Germany) as lipid, and 0.8 mL of Tween 80 (Merck, Darmstadt, Germany) as a surfactant were solved in 10 mL of dichloromethane (Merck, Darmstadt, Germany). Next, thyme oil was mixed manually with the dichloromethane solution. The lipid phase was homogenized with 4% w/v Polyvinyl alcohol (PVA) (Merck, Darmstadt, Germany) (800000 Da) solution using a high-speed sonicator (ultra-probe sonicator, Hielscher UP400s, Germany) at 15000 rpm for 10 min and a white cloudy emulsion containing SLNs was produced. The organic solvents were completely evaporated using a Rotary evaporator at 35°C (Heidolph Hei-VAP Value, Germany) [35].

## 2.3. Preparation of chitosan nanoparticles containing thyme oil

Chitosan solution was prepared by dissolving 1 g of chitosan (9012-76-4, Sigma-Aldrich, Germany) in 100 mL of acetic acid (1% v/v) (Merck, Darmstadt, Germany) using sonication for 70 min. Next, 0.5 mL of tween 80 was added and stirred at 45°C for 90 min, and the pH was adjusted to 4.2. A 1:2 ratio of thyme oil to chitosan solution was created and homogenized in an ice bath for 5 min in order to produce the TO-CH emulsion. Next, to induce chitosan ionic gelation, sodium tripolyphosphate (TPP) (7758-29-4 Sigma-Aldrich) solution (1% w/v) was added with continuous agitation. Subsequently, samples were centrifuged at 6000 g at 4°C for 20 min before being rinsed three times with deionized water [36].

## 2.4. Physicochemical evaluations of nanoparticles

**2.4.1. Particle size, polydispersity index (PDI) measurement, and zeta potential.** Nanoparticles containing thyme oil were evaluated for PDI, particle size, and zeta potential at 25°C using a Zetasizer (Malvern Instrument, UK). Deionized water was used to suspend the samples. Calculations were performed using Zetasizer Ver. 6.01 software with a count rate of 206.3 kcps and a measurement position of 4.65 mm [37].

**2.4.2. Transmission electron microscopy analysis.** The morphology and shape of the thyme oil nanoparticles were studied using Transmission electron microscopy (TEM) (Zeiss-EM10C-100 KV, Germany) at 80 kV. The samples were put on mesh grids, coated with a low Formvar. Next, the grids were subjected to the process of evaporation with the aim of preparing them for TEM analysis. TO-SLN was negatively stained for better visualization [38].

**2.4.3. Fourier Transform Infrared (FTIR) spectrometry.** To examine possible incompatibilities between oils and incorporated excipients in nanoparticles, thyme oil, TO-SLN and TO-CH were assessed using an FTIR (PerkinElmer ES Version 10.5.3, MA, USA). The FTIR spectra of samples were provided from 400–4000 cm$^{-1}$ range with of 4 cm$^{-1}$ resolutions [39].

**2.4.4. % Encapsulation efficiency.** The EE% of nanoparticles was calculated as follows: 500 mg of TO-SLN and TO-CH were first suspended in 10 mL of medium (distilled water and Tween-80 concentration at 2% (v/v)), separately. The suspension was centrifuged at 6000 rpm for 20 min at 25°C. The concentration of free thyme oil from nanoparticles was determined in the supernatant fluid by UV–vis spectrophotometry (Agilent Technologies, Cary 60, Santa Clara, CA, USA) at 285 nm. Then, the encapsulation efficiency was obtained using Eq 1 [35]:

$$\%EE = \frac{WI - W0}{WI} \times 100 \qquad 1$$

WI = primary amount of thyme oil.
W0 = quantity of not loaded thyme oil (free)

## 2.5. The release rate and kinetic of thyme oil from nanoparticles

In order to assess the release rate of thyme oil from nanoparticles, 1 g of each nanoparticle was placed into a 14 kDa dialysis bag (Sigma, Steinheim, Germany), separately and securely sealed on both ends. The bag was placed into the 100 mL receptor medium; distilled water containing tween 80 (2% v/v), at 25°C under constant stirring (100 rpm). Then, 3 mL of samples were collected at 0, 1,3, 6, 24, 48, and 72 h intervals which were substituted with fresh medium to remain the sink condition. Thyme oil amounts were measured using UV spectrophotometry at 285 nm [40].

Several mathematical equations were calculated to assess the kinetics of thyme oil release from TO-CH and TO-SLN. The best curve fit of the release amounts was evaluated with the

mathematical models of Zero-order in addition to first-order, Higuchi, and Korsmeyer-Peppas [41].

## 2.6. Cell viability study

The anti-proliferative effects of the thyme oil and its nanoparticles on HU02 cell lines (Foreskin fibroblast) were assessed utilizing the MTT assay. Simply, cells were seeded into a 96-well culture plate ($5 \times 10^3$ cells mL$^{-1}$) in 100 μl of DMEM medium for 24 h (5% $CO_2$). Next, the medium was replaced with thyme oil, TO-CH, and TO-SLN at a concentration of 5 mg mL$^{-1}$ for 24 h. Subsequently, cell culture plates were incubated with MTT solution (Sigma; 5 mg mL$^{-1}$ of PBS) for 3 hours more at 37˚C. After the removal of the medium, 0.15 mL of Dimethyl Sulfoxide (DMSO) was added to each well. The absorbance was quantified at 570 nm using a microplate spectrophotometer (BioTek Epoch, Santa Clara, CA, USA) (n = 3) [42].

## 2.7. Antibacterial evaluations

**2.7.1. Bacteria and culture conditions.** In current study, the antimicrobial activity of thyme oil, TO-SLN, and TO-CH was investigated against *S. typhimurium* (ATCC 14028) and *E. coli* (ATCC 25922). Bacterial standard stock which had been kept at -30˚C, was cultured on Mueller Hinton Agar (MHA) (Merck, Germany) plates for 24 h at 37˚C.

**2.7.2. Minimum inhibitory/bactericidal concentration determination.** Broth microdilution method was considered for measuring the MIC of thyme oil, TO-SLN, and TO-CH in a 96-well plate, by evaluating the visible growth of microorganisms. In a 96-well sterile plate, the samples were serially diluted using sterile Mueller Hinton Broth (MHB) (Merck, Germany). The positive control consisted of MHB with tested bacteria, and the negative control was pure MHB. Also, chitosan and solid lipid nanoparticles without thyme oil were assessed and compared as control, against both bacterial strains. The certain concentration of microorganisms was set to $10^6$ CFU mL$^{-1}$ for all wells and incubated at 37˚C for 24 h. Then, the plates were visually inspected for turbidity, and the lowermost concentration of added samples that inhibited the growth of bacteria was considered as the MIC value. To accurately evaluate bacterial growth inhibition in serial dilution samples, a blank serial dilution with no bacteria was used as a control. Then the absorbance of each well was determined using an ELISA reader at 640 nm. After determining the MIC, 100 μL of all wells that had shown no visible bacterial growth were sub-cultured on MHA and incubated at a temperature of 37˚C for another 24 h aerobically to determine the MBC (the lowest concentration of a antimicrobial agent which destroys 99.9% of initial bacteria) [43].

## 2.8. TO-SLN and TO-CH for *in vivo* preservation of orange juice

**2.8.1. Preparation of inoculated fruit juice.** Oranges at commercial maturity were purchased from a local market (Akvan, Rasht). After being washed, oranges were peeled off and made into juices. The suspension of *S. typhimurium* was mixed with fruit juice to result in final concentration of $10^3$ CFU/mL and the inoculated juice mixtures were divided into 10 mL sterilized glass vials [44].

**2.8.2. *In vivo* effects of nanoparticles against S. *typhimurium*.** Four orange juice vials were considered containing: 1- TO-SLN, 2- TO-CH, 3- Pure TO and 4- Juice inoculated with bacteria as control. Nanoparticles and oil were added with the concentration equal to MIC, ½ MIC and ¼ MIC. Next, the treated vials were stored at 25˚C up to 6 days and samples were withdrawn on 0, 2nd, 4th, and 6th day [44].

**Table 1. Average size, zeta potential, PDI, and entrapment efficiency of thyme oil loaded solid lipid nanoparticles (TO-SLN) and thyme oil loaded chitosan nanoparticles (TO-CH).**

|   | Sample | Average Size | Zeta Potential | PDI | EE % |
|---|--------|--------------|----------------|-----|------|
| 1 | TO-SLN | 42.47 nm | - 14.4 mV | 0.08 | 73.4±1 |
| 2 | TO-CH | 144.8 nm | +32.3 mV | 0.267 | 81.6±1 |

## 2.9. Statistical analysis

GraphPad Prism 7.0 and SPSS 22.0 software were used for the statistical analysis, which included independent t-tests and one-way analysis of variance (ANOVA). The level of significance was considered at 5% ($p < 0.05$).

## 3. Results

### 3.1 Nanoparticle preparation and evaluations

**3.1.1 Particle size and zeta potential.** The particle size and zeta potential of nanoparticles were recorded experimentally with DLS (Table 1). The result illustrated that the average size of the TO-SLN was about 42.47 nm and it had a surface charge of -14.4 mV with PDI of 0.08 (Fig 1a and 1b). A relatively high zeta potential indicates that they tend to repel each other and exist as a stable dispersion. Whereas, the average size of TO-CH was around 144.8 nm with a PDI of 0.267. The Z-potential values for TO-CH were +32.3 mV (Fig 1c and 1d).

**3.1.2. % Encapsulation efficiency.** The ability of chitosan and SLN to encapsulate thyme oil was evaluated through the EE% determination. The amount of EE obtained was 73.4±1% and 81.6±1% for TO-SLN and TO-CH, respectively. These results of the encapsulation efficiency of nanoparticles proved the suitability of the preparation method. Results of physicochemical characterization of nanoparticles are summarized in Table 1.

**3.1.3. Transmission electron microscopy characterization.** The morphological characteristics of TO-SLN and TO-CH were analyzed using TEM. The TEM depicted that the formed

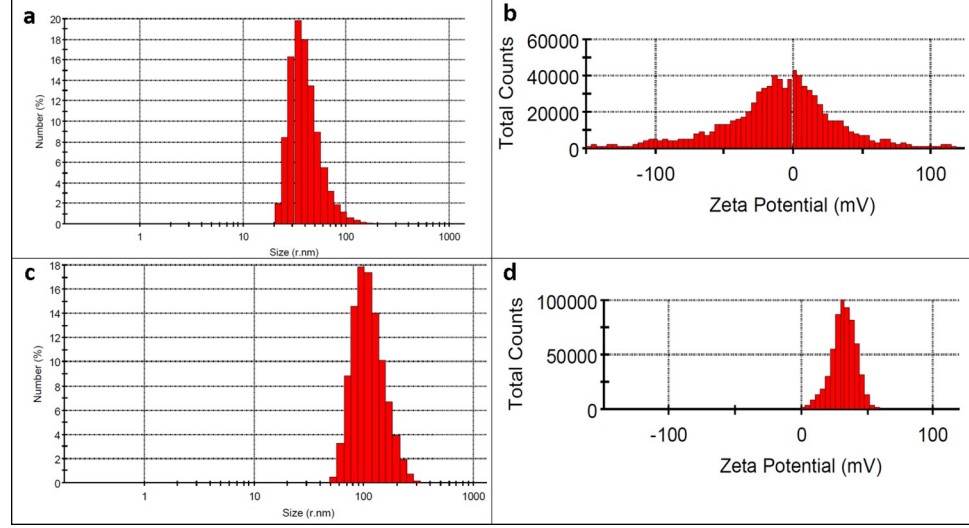

**Fig 1. Data representative for physico-chemical characteristics of thyme oil nanoparticles.** a) hydrodynamic diameter measurement of TO-SLN by DLS. b) zeta potential of TO-SLN. c) hydrodynamic diameter measurement of TO-CH using DLS. d) zeta potential of TO-CH.

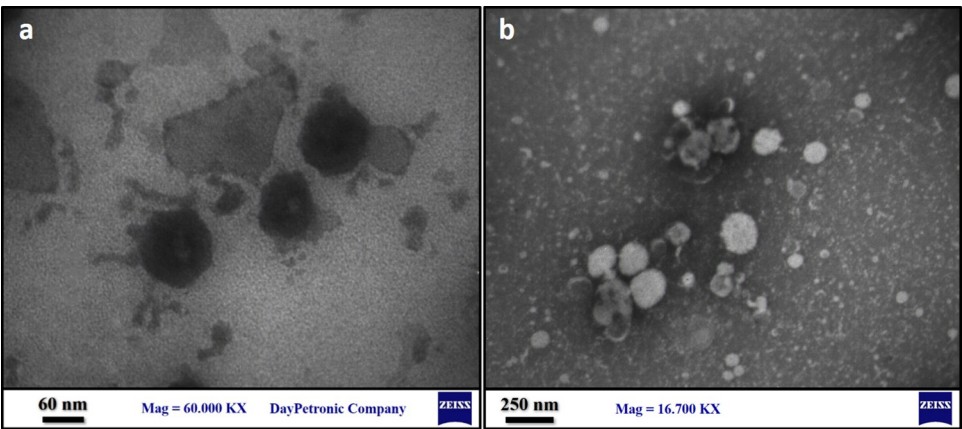

**Fig 2. TEM two-dimensional structural qualification.** a) TO-SLN on the scale of 60 nm, b) TO-CH on the scale of 250 nm.

TO-SLN and TO-CH are spherical/ovoid-shaped with clear edge particles and the particles are well dispersed as well (Fig 2a and 2b).

**3.1.4. Fourier transform infrared spectrometry.** Obtained FTIR results showed that in current study, thyme oil (Fig 3, Blue) revealed the characteristic peaks which were also observed in previous studies, including ones at 3470 cm$^{-1}$ (stretching of O–H in water or alcohol or phenolic structures like thymol), 2961, 2927 and 2872 cm$^{-1}$ assigned to asymmetrical and symmetrical C–H stretching. Also, the strong bands in the range between 1619 cm$^{-1}$ and 1581 cm$^{-1}$ (C = C–C stretching), signal in the range between 1458 and 1380 cm$^{-1}$ (asymmetrical and symmetrical bending of CH$_2$ and CH$_3$), respectively. Various peaks in 1228–943 cm$^{-1}$ and 860–810 cm$^{-1}$ were associated with aromatic C–H in-plane and out-of-plane bending, respectively, confirming the presence of aromatic rings [45, 46]. Aforementioned characteristic peaks are also observable in TO-SLN and TO-CH spectra which confirms the stability of thyme oil in the process of nano-encapsulation.

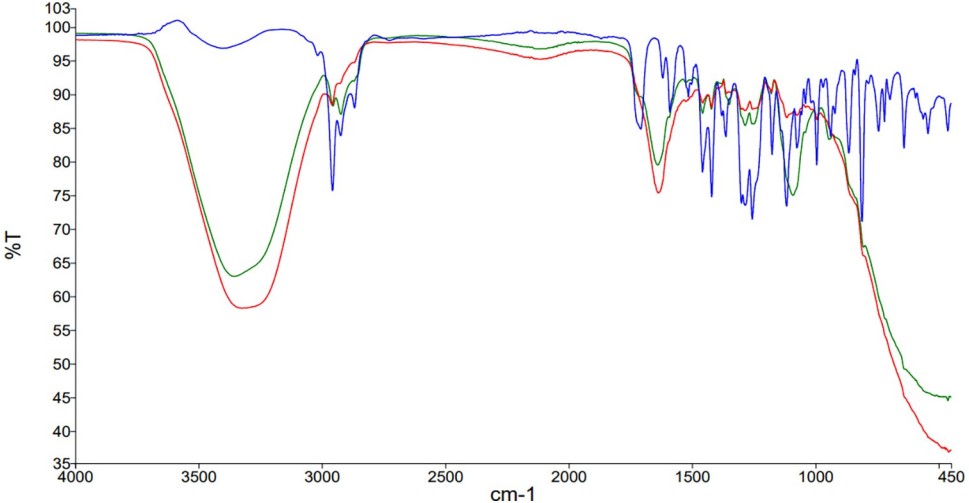

**Fig 3. ATR-FTIR spectra of thyme oil (blue), TO-SLN (green) and TO-CH (red).**

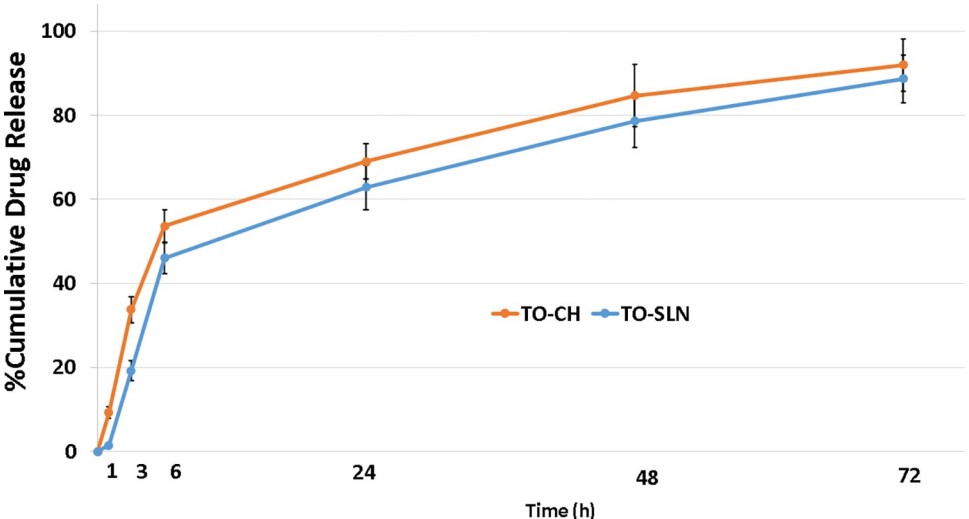

**Fig 4. Data represents the cumulative release profile of thyme oil from TO-SLN and TO-CH in 72 h.**

## 3.2. The release of thyme oil from nanoparticles and kinetics

In this study, the dialysis bag method was used to determine the oil release rate from nanoparticles. The dialysis membrane, retained the nanoparticles and allowed the transfer of the oil immediately into the receiver medium. The cumulative thyme oil release of TO-CH and TO-SLN was analyzed at 285 nm and is shown in Fig 4. TO-SLN and TO-CH had the burst release of 46.03±3.79% and 53.56±3.98% in the first 6 h, respectively. After the burst release, there was a lengthy slow-release duration in which a cumulative release of 88.64±5.64% and 91.95±6.18% in TO-SLN and TO-CH was achieved in 72 h, respectively (S1 and S2 Tables). TO-CH showed a relatively higher release rate than TO-SLN which the difference was not statistically significant ($p>0.05$).

Various release kinetic models including zero order, first order, Higuchi, Korsemeyer-Peppas and Hixson crowell were considered to determine the most matching kinetic model for the formulations based on the highest regression coefficient ($R^2$). Drug release kinetic in TO-CH determined the highest regression coefficient with the Higuchi model ($R^2 = 0.9855$) while TO-SLN revealed the most $R^2$ (0.94) with the Zero order model (S3 Table).

## 3.3. Biocompatibility study

The MTT test was utilized to compare the cell compatibility of TO-SLN and TO-CH with thyme oil on Foreskin fibroblast (HU02). The test was conducted using a 5 mg mL$^{-1}$ concentration of both nanoparticles and pure thyme oil. Results revealed a higher than 70% cell viability for all three samples. However, the cell viability for thyme oil (84.58±4.27) was higher than that of TO-SLN (72.5±2.35) and TO-CH (80±2.89). The difference was not statistically significant ($p>0.05$), hence, both nanoparticles and thyme oil were considered cell-compatible materials (Fig 5).

## 3.4. Determination of minimum inhibitory/bactericidal concentration

The MIC and MBC results of thyme oil, TO-SLN, and TO-CH were obtained against *E. coli* and *S. typhimurium* using the broth microdilution test, as presented in Table 2. The results of the experiment revealed discernible antimicrobial effects against the microorganisms under

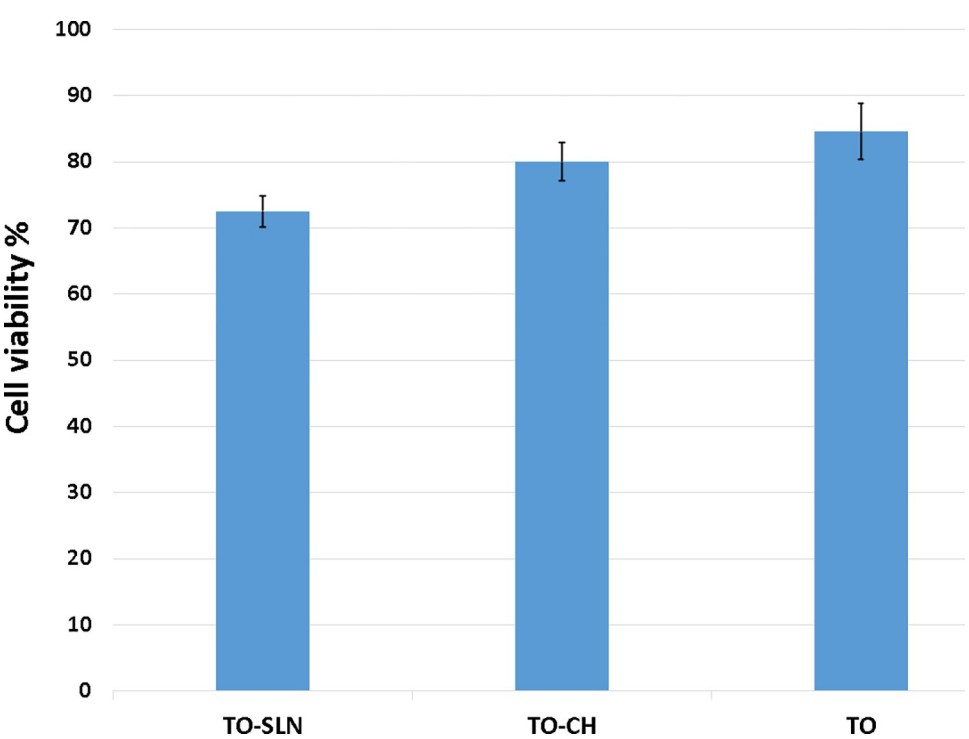

**Fig 5. The cell viability% treated with thyme oil (mg mL$^{-1}$) compared with thyme oil loaded SLN and chitosan nanoparticles (5 mg mL$^{-1}$).**

investigation. *S. typhimurium* was observed to have a higher susceptibility to thyme oil in comparison to *E. coli*. About *S. typhimurium*, the obtained MICs for thyme oil, TO-SLN, and TO-CH were 150, 40, and 1.5 µg mL$^{-1}$, and MBC were 290, 160, and 45 µg mL$^{-1}$, respectively. Also, against *E. coli*, the MICs were found to be 180, 60, and 4 µg mL$^{-1}$, and the concentrations of 310, 190, and 35 µg mL$^{-1}$ were found to be the MBC, respectively. Notably, controls; solid lipid nanoparticles without thyme oil did not show any antibacterial effect against either strain and chitosan nanoparticles without thyme oil showed a weak imperceptible effect. The result illustrated that thyme oil had the lowest antibacterial activity against both microorganisms compared to nanoparticles. Although the comparison between the MIC and MBC ranges revealed that SLN potentiates the antimicrobial effect of thyme oil against microorganisms, a noteworthy enhancement was shown in the antimicrobial effect of TO-CH as a carrier system.

### 3.5. I*n vivo* preservation of orange juice by TO-SLN and TO-CH

Since Thyme oil and its nanoparticles were able to kill food spoilage bacteria in *in vitro* tests, its activity in a real food system (orange juice) has also been studied. Due to the lower MIC and MBC presented against *S. typhimurium*, it was considered for the *in vivo* test. reduction in

**Table 2. The MIC and MBC values (µg mL$^{-1}$) for pure thyme oil, thyme oil loaded solid lipid nanoparticles (TO-SLN) and thyme oil loaded chitosan nanoparticles (TO-CH) against *S. typhimurium* and *E. coli*.**

| | Thyme oil | | TO-SLN | | SLN | | TO-CH | | Chitosan | |
|---|---|---|---|---|---|---|---|---|---|---|
| | MIC | MBC | MIC | MBC | MIC | MBC | MIC | MBC | MIC | MBC |
| *E. coli* (ATCC 25922) | 180 | 310 | 60 | 190 | - | - | 4 | 35 | 29600 | 47000 |
| *S. typhimurium* (ATCC 14028) | 150 | 290 | 40 | 160 | - | - | 1.5 | 25 | 29800 | 47000 |

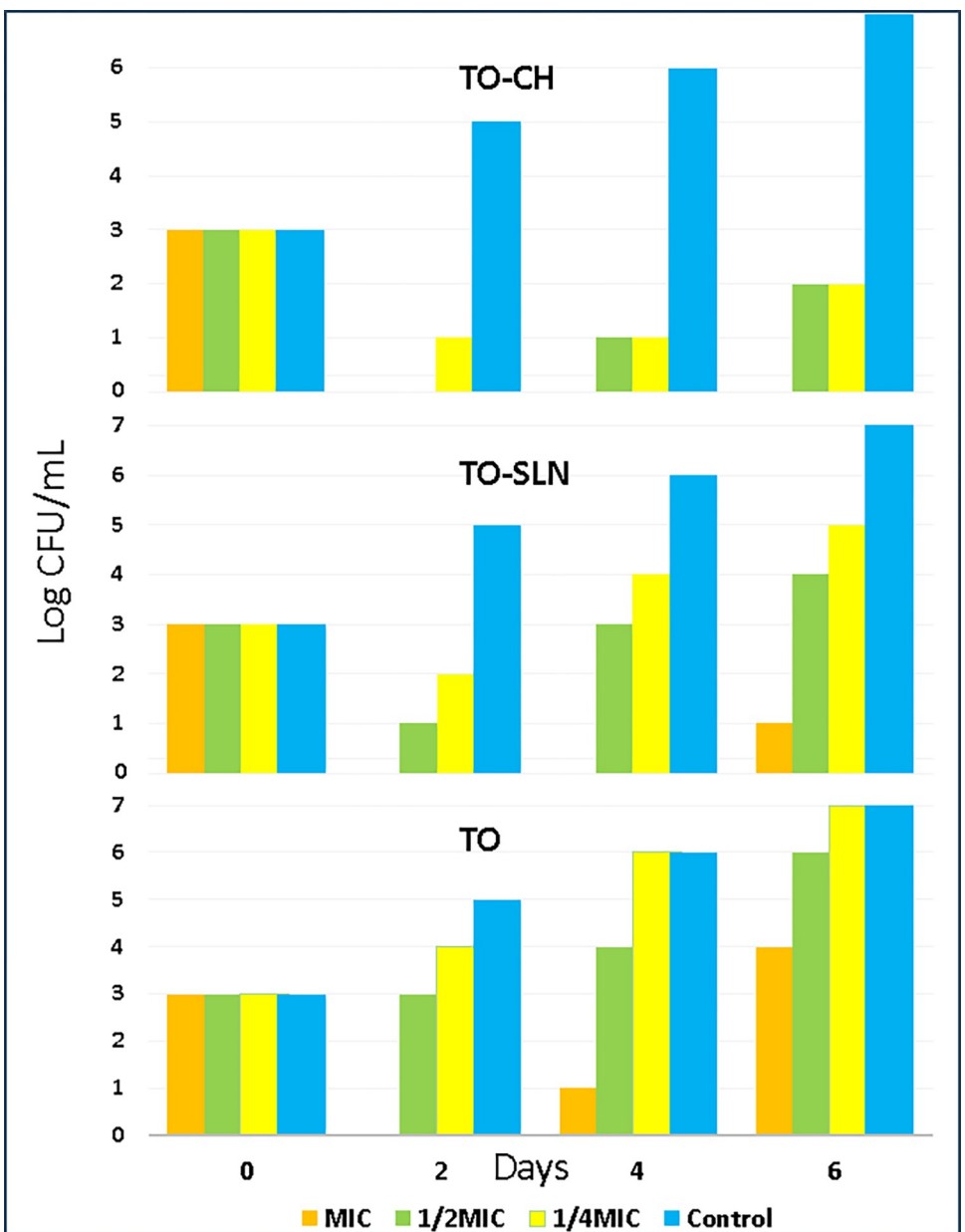

**Fig 6. Variation of *S. typhimurium* viability in presence of TO, TO-SLN and TO-CH with the concentrations of MIC, ½ MIC and ¼ MIC in days 0,2,4 and 6.**

viability of *S. typhimurium* due to thyme oil, TO-SLN and TO-CH treatment in dose-dependent (MIC, 1/2 MIC, and 1/4 MIC) and time-dependent manner (i.e., 0, 2, 4, and 6 days) was evaluated. As is presented in Fig 6, the *in vivo* antibacterial effects showed a dose-dependent but not a time-dependent manner. The highest growth prevention was achieved by TO-CH at the MIC concentration with no growth until 6[th] day and TO with ¼ MIC did not show any significant difference with control (containing no antimicrobial agent) in days 4 and 6 (S4 Table). Colony growth in day 6 for all samples is presented in S1 Fig.

## 4. Discussion

Within the realm of the food industry, ensuring food safety and preventing cross-contamination of pathogens such as *S. typhimurium* and *E. coli* are of major concerns [47–49]. Despite significant advancements in food preservation and safety methods, major outbreaks of foodborne pathogen-related diseases represent a critical public health problem [50]. Nowadays, the increasing concerns about using artificial food additives have turned the attention of researchers to EOs as antimicrobial and preservative substances of natural origin against pathogenic microorganisms and food spoilage, and biofilm-forming bacterial strains [51]. Hence, the ability of thyme oil to prevent the biofilm formation and growth of *E. coli* and *S. typhimurium* is well-documented for its antimicrobial and antioxidant properties [52].

Beside the high potential of EOs to inhibit the activity of microorganisms, their chemical instability and susceptibility to oxidation and volatilization limit the uses of EOs in food preservation [53, 54]. Nanotechnology has been used as a powerful means to improve food shelf life, nutrient delivery, enhanced tastes and flavors and develop nano-based choices to barricade food spoilage [55]. Recently, the addition of nanoparticles to packaged foods as antibacterial agents has shown promising results [56, 57]. Notably, various types of research have been conducted to create an ideal antibacterial and antibiofilm agent, including thyme oil to hamper the activity of bacteria causing foodborne illnesses [27, 58–60]. However, the antibacterial effects of TO-SLN compared to TO-CH have not been evaluated yet. In this article, we tested the use of nanoparticle-based materials containing thyme oil to inhibit the bacteria responsible for foodborne illnesses and spoilage, and chitosan nanoparticles were assessed in comparison to SLN as a delivery system for thyme oil.

Different nanocarriers have been developed for delivery of thyme constituents. Among physicochemical characteristics, Moghimipour et al. in 2023 designed a Thymol-loaded SLNs using a microemulsion metho for colon cancer treatment. Prepared SLNs had the average particle size of 145 nm and EE% was 63% [61]. However, the size of nanoparticles was similar with current study (144.8 nm), the EE% was significantly lower (73.4% in current study) which can be justified by the difference of preparation methods. Since, EE% is a factor which is strongly affected by preparation method. Naseri et al. in 2020 formulated SLN containing *Zataria multiflora* EO using high-pressure homogenization technique in which all formulations showed a higher than 200 nm particle size and higher than 85% of EE. The particle size and EE% was higher compared to our study, probably due to difference in preparation technique [62]. Considering that however increase in EE is desirable, decrease in size for clinical applications is mainly of interest. Perez et al. in 2022, designed a nanostructured archaeolipid carriers (NAC) containing *Thymus vulgaris* essential oil (EOT) and tobramycin by emulsion-ultrasonication method. Results showed that the size of NAC was 241 nm while the EE of EOT could not be determined [63]. It seems that however the preparation method was similar in both studies, concomitant loading of two active ingredient leads to a greater particle size. Granata et al. in 2021, encapsulated Oregano and Thyme (Th-NCP) EOs in chitosan nanoparticles, using ionotropic gelation. For h-NCP, the main peak for particle size appeared in 449 nm with EE of 80% [27]. Although, the fabrication method and EE% is similar to current study, the particle size is significantly higher (compared to 42.48 nm) which may be due to probe sonication process. Hosseini et al. in 2018 developed thyme essential oil loaded in chitosan nanoparticles using ionotropic gelation, that size of particles ranged between 30–100 nm with the EE of 63.8% [64]. However, the size can be considered in the same range with current results, the EE% is significantly lower.

Several researches have investigated the antimicrobial effects of thyme oil against bacteria responsible for food spoilage, including *S. typhimurium* and *E. coli* [65–67]. In our study,

obtained MIC and MBC for thyme oil were 180 μg mL$^{-1}$and 310 μg mL$^{-1}$, 150 μg mL$^{-1}$and 290 μgml$^{-1}$ μg mL$^{-1}$ against *E. coli* and *S. typhimurium*, respectively, which showed a decrease compared to the study of Al Hafi et al. in 2017 (MIC ranging from 0.1 to 0.8 mg mL$^{-1}$, six pathogenic bacteria (*S. aureus*, *E. coli*, *E. faecalis*, *P. aeruginosa*, *S. enterica*, and *Bacillus subtilis*)) [65]. Also, there are other studies that reported a higher MIC value for thyme oil, against *salmonella*, including 0.6 mg mL$^{-1}$ [68] and 1 mg mL$^{-1}$ of thyme oil [52]. Moreover, Altiok et al. in 2010, indicated that the minimum concentration of thyme oil that inhibits the growth of *E. coli* was 1.2% (v/v), which was much higher than our results [69]. Evaluating the MIC of thyme oil and TO-SLN in this study showed the MIC for TO-SLN significantly diminished around 60 and 40 μg mL$^{-1}$for *E. coli* and *S. typhimurium*, respectively ($p < 0.05$). Similarly, Reis Mockdeci et al. in 2023, reported that tea tree oil loaded SLN presented fungicidal activity for *C. guilliermondii* (Minimum fungicidal concentration (MFC): 5.6 mg mL$^{-1}$) and other species (MFC: 11.20 mg mL$^{-1}$), whereas regarding tea tree oil, the fungicidal effect was only observed for *C. albicans* (MFC: 25 mg mL$^{-1}$) [70]. Likewise, Nemattalab et al. in 2022, encapsulated cinnamon oil into solid lipid nanoparticles, which showed a notable decline in the MIC of cinnamon oil-loaded SLN against *E. coli* in comparison to pure cinnamon oil, since the MIC declined from 155–165 μg mL$^{-1}$for cinnamon oil to 60–75 μg mL$^{-1}$for cinnamon oil-loaded SLN [42]. In justifying the relatively higher influence of the thyme oil nanoparticles in comparison with the pure thyme oil, SLNs can inhibit the evaporation of the essential oil and increase its stability and duration of action. Actually, several investigations indicated the suitability of SLNs as carriers for EO, and some of the EOs have been prominently incorporated into SLNs [71, 72].

Also, in the current study, results revealed that in comparison with TO-SLN, TO-CH impressively diminished the MIC and MBC to 35 and 25 μg mL$^{-1}$and 4 and 1.5 μg mL$^{-1}$for *S. typhimurium* and *E. coli*, respectively. In similar studies, Granata et al. in 2021, illustrated that chitosan nanoparticle containing thyme oil increased bactericidal effects against *S. aureus*, *E. coli*, and *L. monocytogenes* compared to pure oil. For thyme-chitosan nanoparticles, MIC and MBC amounts were less (four to seven times) than thyme oil [27]. Moreover, in a similar study conducted on cinnamon oil, Rohani et al. in 2023, declared that CO-CH showed six times stronger antimicrobial activity compared to CO-SLN and more than ten times higher effects compared to cinnamon oil [73].

The higher antibacterial effects of TO-CH can be justified by inherent antimicrobial activity of chitosan than can be synergized along with other antimicrobial agents. Various mechanisms have been proposed for antimicrobial potency of chitosan including the polycationic charge of chitosan (due to amino groups originated from deacetylation of glucosamine monomer in chitin structure) that interferes with the cell membrane's anionic components, lead the chitosan nanoparticles as active carrier which facilitate the transport of EO through the bacterial cell membrane [74]. Also, chitosan interacts with the membrane of the cell to alter cell permeability, decreasing the bacterial survival [75]. The other mechanism of low Mw chitosan involves the binding of chitosan with DNA to inhibit RNA synthesis [76]. Flocculation of electronegative elements by CS in the cell upsets the physiological activities of the bacteria and leads to bacterial cell death [77].

These results emphasized the significant antimicrobial activity of TO nanoparticles, especially TO-CH, compared with TOS-LN and thyme oil, making them interesting candidates as natural food preservatives. These eco-friendly nano-systems have the potential to improve food safety and serve as a strong substitute for artificial preservatives. Hence, the encapsulation of thyme oil in chitosan nanoparticles is a promising trajectory for their new application in pharmaceutical and food products.

## 5. Conclusion

In conclusion, thyme oil is a promising candidate for the application as natural preservative for food products, especially when encapsulates in chitosan as nano-carrier. TO-CH presented 10-fold higher antibacterial activity compared to pure thyme oil and 6-fold stronger effect in comparison with TO-SLN. Current results showed strong potentiation of antibacterial effects along with reputable physico-chemical characteristics of TO-CH including the average particle size: 144.8 with PDI: 0.2, EE%: 81.6 and zeta potential: +32.3 mV. Further evaluations of TO-CH effects on food contaminating fungal species would be advantageous. Also, the scale up of TO-CH production and long-term stability studies should be considered for practical application of this natural preservative in food industry.

## Supporting information

**S1 Table. Absorption, concentration and cumulative oil release from TO-SLN in 72 h.**
(DOCX)

**S2 Table. Absorption, concentration and cumulative oil release from TO-CH in 72 h.**
(DOCX)

**S3 Table. Release kinetic parameters for TO-SLN and TO-CH based on mathematical models.**
(DOCX)

**S4 Table. Log concentrations of inoculated *S. typhimurium* in fresh orange juice in presence of TO, TO-SLN and TO-CH in 6 days.**
(DOCX)

**S1 Fig. Culture plates for counting the CFU existed in orange juice in day 6 of the *in vivo* study.** a) Pure oil with MIC concentration. b) TO-SLN with MIC concentration. c) TO-CH with MIC concentration. d) control containing no antimicrobial agent.
(DOCX)

## Acknowledgments

The Registering institute of this research is Guilan University of Medical Sciences under the number of 5475 which is acknowledged.

## Author Contributions

**Conceptualization:** Shirin Parvinroo, Zahra Hesari.

**Data curation:** Amirhosein Shabgoo Monsef, Mehran Nemattalab.

**Formal analysis:** Mehran Nemattalab.

**Methodology:** Amirhosein Shabgoo Monsef, Mehran Nemattalab, Zahra Hesari.

**Project administration:** Zahra Hesari.

**Supervision:** Zahra Hesari.

**Visualization:** Zahra Hesari.

**Writing – original draft:** Amirhosein Shabgoo Monsef, Mehran Nemattalab.

**Writing – review & editing:** Zahra Hesari.

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
