## [Decision Letter · Decision Letter 0]

16 Sep 2024

PONE-D-24-35651Antibacterial effects of thyme oil loaded Solid Lipid and chitosan nano-carriers against Salmonella Typhimurium and Escherichia coli as food preservativesPLOS ONE

Dear Dr. Hesari,

Thank you for submitting your manuscript to PLOS ONE. After careful consideration, we feel that it has merit but does not fully meet PLOS ONE’s publication criteria as it currently stands. Therefore, we invite you to submit a revised version of the manuscript that addresses the points raised during the review process.

The manuscript entitled “Antibacterial effects of thyme oil loaded Solid Lipid and chitosan nano-carriers against Salmonella Typhimurium and Escherichia coli as food preservatives” is original and good concept of sustainable delivery of thyme oil as an antimicrobial agents for applications in food and pharma sector.

The manuscript is novel and showed scientific sound. Therefore, the manuscript must be revised according to the comments of reviewers. The manuscript shall be reconsider after corrections. 

We look forward to receiving your revised manuscript.

Kind regards,

Nishant Kumar, Ph.D

Academic Editor

PLOS ONE

A clean copy of the edited manuscript (uploaded as the new *manuscript* file).

3. We note that your Data Availability Statement is currently as follows: [All relevant data are within the manuscript and its supporting information files.] Please confirm at this time whether or not your submission contains all raw data required to replicate the results of your study. Authors must share the “minimal data set” for their submission. PLOS defines the minimal data set to consist of the data required to replicate all study findings reported in the article, as well as related metadata and methods (https://journals.plos.org/plosone/s/data-availability#loc-minimal-data-set-definition). For example, authors should submit the following data: - The values behind the means, standard deviations and other measures reported; - The values used to build graphs; - The points extracted from images for analysis. Authors do not need to submit their entire data set if only a portion of the data was used in the reported study. If your submission does not contain these data, please either upload them as Supporting Information files or deposit them to a stable, public repository and provide us with the relevant URLs, DOIs, or accession numbers. For a list of recommended repositories, please see https://journals.plos.org/plosone/s/recommended-repositories. If there are ethical or legal restrictions on sharing a de-identified data set, please explain them in detail (e.g., data contain potentially sensitive information, data are owned by a third-party organization, etc.) and who has imposed them (e.g., an ethics committee). Please also provide contact information for a data access committee, ethics committee, or other institutional body to which data requests may be sent. If data are owned by a third party, please indicate how others may request data access.

Additional Editor Comments:

Dear Authors,

The manuscript entitled “Antibacterial effects of thyme oil loaded Solid Lipid and chitosan nano-carriers against Salmonella Typhimurium and Escherichia coli as food preservatives” is original and good concept of sustainable delivery of thyme oil as an antimicrobial agents for applications in food and pharma sector.

The manuscript required major improvement. My comments and suggestions are appended below.

Abstract

Objective and conclusion must be clear

Introduction

Introduction must be improving with focusing on the nanoparticles and thyme oils

Some previous studies finding should be highlights in the introduction section

Results and discussion

The results must be elobrate in more depth and discussion section must be improve

Conclusion

Conclusion section must be rewrite with highlights the findings of the study

Reviewers' comments:

Reviewer's Responses to Questions

**Comments to the Author**

1. Is the manuscript technically sound, and do the data support the conclusions?

Reviewer #1: Yes

Reviewer #2: Partly

2. Has the statistical analysis been performed appropriately and rigorously? 

Reviewer #1: Yes

Reviewer #2: N/A

3. Have the authors made all data underlying the findings in their manuscript fully available?

Reviewer #1: Yes

Reviewer #2: Yes

4. Is the manuscript presented in an intelligible fashion and written in standard English?

Reviewer #1: Yes

Reviewer #2: No

5. Review Comments to the Author

Reviewer #1: The authors have investigated the antibacterial effects of thyme oil loaded solid lipid and chitosan nano-carriers against Salmonella Typhimurium and Escherichia coli as food preservatives. The study has highlighted important parameters which support the findings. The data is well presented by the authors. Some minor revisions are recommended for further improvements in the manuscript:

1. In section 2.1, mention the full form of abbreviation PVA.

2. In section 2.3.1, 2.3.2, 2.3.4 insert the references for the methodology followed.

3. Mention the abbreviations of samples in the legend of table 1, table 2

4. In the section 4, the year of publication should be mentioned with the references ‘Reis Mockdeci et al.’, Nemattalab et al., Al Hafi et al etc.

5. The section 4 (discussion) could be improved by stating systematic explanation of the outcomes of each result along with the comparison with similar studies.

6. A single format of writing (p < 0.05) should be used uniformly throughout the manuscript. In section 4, its mentioned as (Pvalue < 0.05) which differs from rest of the manuscript.

7. The conclusion section could be written in a more scientific manner by explaining the pros of the outcomes and future scope of the study.

8. Maintain same font style and font size as recommended in the author guidelines. The font style and font size are different for references than rest of the manuscript.

Reviewer #2: The manuscript titled as Antibacterial effects of thyme oil loaded Solid Lipid and chitosan nano-carriers against Salmonella Typhimurium and Escherichia coli as food preservatives submitted to PLOS ONE with manuscript number PONE-D-24-35651 for possible publication is not suitable for the publication as such due to following points. It must be REJECTED.

• The antibacterial results are limited to in vitro experiments. The study could be strengthened by including in vivo evaluations to confirm its potential as a food preservative.

• While the study compares TO-SLN and TO-CH, it doesn't explore why chitosan significantly enhances antibacterial effects compared to lipid-based systems. A deeper analysis of the underlying mechanisms would be valuable.

• Long-term stability of the nanoparticle systems (especially for food applications) is not addressed, which is essential for practical implementation in the food industry.

• The diagrams need to be improved.

• Some sentences are lengthy and complex, which may obscure the key points so simplifying those sentences would improve clarity.

• Discussion section is comprehensive and the comparison with existing literature is good but it can be improved by adding more information on how TO-CH nanoparticles achieve such high antibacterial activity and how and why is it more effective than others.

• Ensure the following: Avoid repetition of words, The terms in vitro, in vivo, names of the biological organisms should be italic.

• Conclusion is very concise and summarizes only the findings of this study so emphasizing on the practical application of TO-CH as a natural food preservative and addressing the limitations of this study could strengthen the conclusion and make it more impactful.

6. PLOS authors have the option to publish the peer review history of their article (what does this mean?). If published, this will include your full peer review and any attached files.

Reviewer #1: No

Reviewer #2: No

---

## [Author Response · Author response to Decision Letter 0]

23 Oct 2024

Dear Editor in chief and Editorial Board of PLOS ONE journal

Thank you for considering our manuscript “PONE-D-24-35651 Antibacterial effects of thyme oil loaded Solid Lipid and chitosan nano-carriers against Salmonella Typhimurium and Escherichia coli as food preservatives” for peer review and thanks to reviewers for their valuable time and comments. All comments are carefully addressed as follows. Due to the comments which needed some additional lab experiments, it took a relatively long time for us to do the revision, however, fortunately we didn’t pass the deadline.

Any further comments and questions are welcome by the research team.

Regards

Zahra Hesari 

Abstract

Objective and conclusion must be clear

Authors: edited and highlighted.

Introduction

Introduction must be improving with focusing on the nanoparticles and thyme oils

Some previous studies finding should be highlights in the introduction section

Authors: some previous studies focusing of thyme oil nanoparticles (chitosan and lipid nanoparticles) has been added and highlighted in introduction. 

Results and discussion

The results must be elobrate in more depth and discussion section must be improve

Authors: edited and highlighted. Results are discussed in a more depth in discussion section.

Conclusion

Conclusion section must be rewrite with highlights the findings of the study

Authors: conclusion has been completely revised.

Reviewer #1: The authors have investigated the antibacterial effects of thyme oil loaded solid lipid and chitosan nano-carriers against Salmonella Typhimurium and Escherichia coli as food preservatives. The study has highlighted important parameters which support the findings. The data is well presented by the authors. Some minor revisions are recommended for further improvements in the manuscript:

1. In section 2.1, mention the full form of abbreviation PVA.

Authors: Mentioned and highlighted.

2. In section 2.3.1, 2.3.2, 2.3.4 insert the references for the methodology followed.

Authors: References added and highlighted.

3. Mention the abbreviations of samples in the legend of table 1, table 2

Authors: Full description of sample names is added and highlighted to the abbreviations which were written before.

4. In the section 4, the year of publication should be mentioned with the references ‘Reis Mockdeci et al.’, Nemattalab et al., Al Hafi et al etc.

Authors: The year of publication is mentioned and highlighted.

5. The section 4 (discussion) could be improved by stating systematic explanation of the outcomes of each result along with the comparison with similar studies.

Authors: systematic explanation for results of physicochemical evaluations of nanoparticles (SLN and chitosan) have been added to discussion and highlighted. 

6. A single format of writing (p < 0.05) should be used uniformly throughout the manuscript. In section 4, its mentioned as (Pvalue < 0.05) which differs from rest of the manuscript.

Authors: all (p ˂0.05) have been assimilated and highlighted. 

7. The conclusion section could be written in a more scientific manner by explaining the pros of the outcomes and future scope of the study.

Authors: conclusion is completely revised based on practical application of TO-CH as food natural preservative and suggestions for future studies are mentioned. 

8. Maintain same font style and font size as recommended in the author guidelines. The font style and font size are different for references than rest of the manuscript.

Authors: the font style and size of references has been coordinated with the rest of manuscript and highlighted.

Reviewer #2: The manuscript titled as Antibacterial effects of thyme oil loaded Solid Lipid and chitosan nano-carriers against Salmonella Typhimurium and Escherichia coli as food preservatives submitted to PLOS ONE with manuscript number PONE-D-24-35651 for possible publication is not suitable for the publication as such due to following points. It must be REJECTED.

• The antibacterial results are limited to in vitro experiments. The study could be strengthened by including in vivo evaluations to confirm its potential as a food preservative.

Authors: Based on referee’s comment The in vivo test was designed based on literature and performed and results are presented under the section of 3.5 and Fig.6. 

• While the study compares TO-SLN and TO-CH, it doesn't explore why chitosan significantly enhances antibacterial effects compared to lipid-based systems. A deeper analysis of the underlying mechanisms would be valuable.

Authors: Added in discussion and highlighted.

• Long-term stability of the nanoparticle systems (especially for food applications) is not addressed, which is essential for practical implementation in the food industry.

Authors: As the long-term stability study needs at least 2 years (based on United State Pharmacopeia) of time, however we have started the stability studies for this and similar projects, the results will be used in scale up and production process. 

• The diagrams need to be improved.

Authors: Diagrams in figures 1, 4 and 5 have been improved among visuality and resolution. 

• Some sentences are lengthy and complex, which may obscure the key points so simplifying those sentences would improve clarity.

Authors: all the manuscript text was revised and edited sentences are highlighted. 

• Discussion section is comprehensive and the comparison with existing literature is good but it can be improved by adding more information on how TO-CH nanoparticles achieve such high antibacterial activity and how and why is it more effective than others.

Authors: added and highlighted.

• Ensure the following: Avoid repetition of words, The terms in vitro, in vivo, names of the biological organisms should be italic.

Authors: all the manuscript text was revised based on this comment, all scientific names of the biological organisms (plants or bacteria) are provided in italic and highlighted. Also, the phrase “in vitro” which was mentioned once. Repetitions are deleted. 

• Conclusion is very concise and summarizes only the findings of this study so emphasizing on the practical application of TO-CH as a natural food preservative and addressing the limitations of this study could strengthen the conclusion and make it more impactful.

Authors: conclusion is completely revised based on practical application of TO-CH as food natural preservative and limitations of current study is mentioned in final part of conclusion as suggestions.

---

## [Decision Letter · Decision Letter 1]

8 Nov 2024

PONE-D-24-35651R1Antibacterial effects of thyme oil loaded Solid Lipid and chitosan nano-carriers against Salmonella Typhimurium and Escherichia coli as food preservativesPLOS ONE

Dear Dr. Hesari,

Thank you for submitting your manuscript to PLOS ONE. After careful consideration, we feel that it has merit but does not fully meet PLOS ONE’s publication criteria as it currently stands. Therefore, we invite you to submit a revised version of the manuscript that addresses the points raised during the review process.

We look forward to receiving your revised manuscript.

Kind regards,

Nishant Kumar, Ph.D

Academic Editor

PLOS ONE

Journal Requirements:

Additional Editor Comments:

Dear Authors,

Your manuscript entitled "Antibacterial effects of thyme oil loaded Solid Lipid and chitosan nano-carriers against Salmonella Typhimurium and Escherichia coli as food preservatives

" could be accepting after some minor corrections.

Reviewer comments:

The revised manuscript "Antibacterial effects of thyme oil loaded Solid Lipid and chitosan nano-carriers against Salmonella Typhimurium and Escherichia coli as food preservatives with manuscript No. PONE-D-24-35651R1 has been reviewed and the authors are appreciated for the revision. However, some minor comments include the

1. It will be better if the equations used in the manuscript are numbered.

2. FTIR data can be plotted in one frame instead of using 3 frames...This will improve the quality of figure.

3. Figure 6 can also be improved by reducing to one frame instead of 3, if possible and data allows.

Reviewers' comments:

Reviewer's Responses to Questions

**Comments to the Author**

1. If the authors have adequately addressed your comments raised in a previous round of review and you feel that this manuscript is now acceptable for publication, you may indicate that here to bypass the “Comments to the Author” section, enter your conflict of interest statement in the “Confidential to Editor” section, and submit your "Accept" recommendation.

Reviewer #2: All comments have been addressed

2. Is the manuscript technically sound, and do the data support the conclusions?

Reviewer #2: Yes

3. Has the statistical analysis been performed appropriately and rigorously? 

Reviewer #2: N/A

4. Have the authors made all data underlying the findings in their manuscript fully available?

Reviewer #2: Yes

5. Is the manuscript presented in an intelligible fashion and written in standard English?

Reviewer #2: Yes

6. Review Comments to the Author

Reviewer #2: The revised manuscript "Antibacterial effects of thyme oil loaded Solid Lipid and chitosan nano-carriers against Salmonella Typhimurium and Escherichia coli as food preservatives with manuscript No. PONE-D-24-35651R1 has been reviewed and the authors are appreciated for the revision. However, some minor comments include the

1. It will be better if the equations used in the manuscript are numbered.

2. FTIR data can be plotted in one frame instead of using 3 frames...This will improve the quality of figure.

3. Figure 6 can also be improved by reducing to one frame instead of 3, if possible and data allows.

7. PLOS authors have the option to publish the peer review history of their article (what does this mean?). If published, this will include your full peer review and any attached files.

Reviewer #2: No

---

## [Author Response · Author response to Decision Letter 1]

18 Nov 2024

Dear Editor in chief and Editorial Board of PLOS ONE journal

Thank you for considering our manuscript “PONE-D-24-35651 Antibacterial effects of thyme oil loaded Solid Lipid and chitosan nano-carriers against Salmonella Typhimurium and Escherichia coli as food preservatives” for second round of peer review and thanks to reviewers for their valuable time and comments. All comments are carefully addressed as follows. 

Any further comments and questions are welcome by the research team.

Regards

Zahra Hesari 

1. It will be better if the equations used in the manuscript are numbered.

Authors: the equation for calculating EE% in section 2.3.4 is numbered and highlighted in green. 

2. FTIR data can be plotted in one frame instead of using 3 frames...This will improve the quality of figure.

Authors: Thanks to valuable comment, FTIR spectrum are provided in one frame as overlay format in figure 3. The figure legend is also edited and highlighted in green. 

3. Figure 6 can also be improved by reducing to one frame instead of 3, if possible and data allows.

Authors: consulting our statistician, we found that if we want to provide all data in one frame, due to presence of numerous variables like 3 samples, 1 control, 4 different concentrations and 4 time points, will make the chart very confusing and the figure caption needs to be too long for explaining numerous colors which leads to a hard to understand figure for readers.

---

## [Editor Report · Decision Letter 2]

27 Nov 2024

Antibacterial effects of thyme oil loaded Solid Lipid and chitosan nano-carriers against Salmonella Typhimurium and Escherichia coli as food preservatives

PONE-D-24-35651R2

Dear Dr. Hesari,

We’re pleased to inform you that your manuscript has been judged scientifically suitable for publication and will be formally accepted for publication once it meets all outstanding technical requirements.

Kind regards,

Nishant Kumar, Ph.D

Academic Editor

PLOS ONE
---

## [Editor Report · Acceptance letter]

16 Dec 2024

PONE-D-24-35651R2 

PLOS ONE

Dear Dr. Hesari, 

I'm pleased to inform you that your manuscript has been deemed suitable for publication in PLOS ONE. Congratulations! Your manuscript is now being handed over to our production team.

Kind regards, 

on behalf of

Dr. Nishant Kumar 

Academic Editor

PLOS ONE